# CRUSTY: a versatile web platform for the rapid analysis and visualization of high-dimensional flow cytometry data

Simone Puccio [1,2,6] ✉, Giorgio Grillo [3,6], Giorgia Alvisi[1], Caterina Scirgolea [1], Giovanni Galletti [1,5], Emilia Maria Cristina Mazza [1], Arianna Consiglio[3], Gabriele De Simone[4], Flavio Licciulli [3] & Enrico Lugli [1] ✉

Flow cytometry (FCM) can investigate dozens of parameters from millions of cells and hundreds of specimens in a short time and at a reasonable cost, but the amount of data that is generated is considerable. Computational approaches are useful to identify novel subpopulations and molecular biomarkers, but generally require deep expertize in bioinformatics and the use of different platforms. To overcome these limitations, we introduce CRUSTY, an interactive, user-friendly webtool incorporating the most popular algorithms for FCM data analysis, and capable of visualizing graphical and tabular results and automatically generating publication-quality figures within minutes. CRUSTY also hosts an interactive interface for the exploration of results in real time. Thus, CRUSTY enables a large number of users to mine complex datasets and reduce the time required for data exploration and interpretation. CRUSTY is accessible at https://crusty.humanitas.it/.

The recent progress in flow cytometry (FCM) technologies and fluorescently-conjugated antibody production have increased the number of parameters being measured at the level of single cells, allowing more researchers to dissect cellular heterogeneity or to detect rare cell populations[1–5]. Currently, traditional flow cytometry can detect up to 30 parameters simultaneously[6], and spectral unmixing can extend measurements to >40[7]. The versatility of the technology allows the investigation of several samples in a variety of experimental conditions and at a reasonable cost, thereby resulting in the generation of large multidimensional datasets, the analysis of which is time-consuming if approached by traditional gating[6,8,9]. Moreover, despite the effort to standardize and harmonize manual gating strategies, the identification of subpopulations through visual inspection is still very common and represents the largest source of variability, thereby impacting reproducibility[10]. Additionally, manual gating is not easily scalable due to the number of biaxial plots and bar graphs which increase exponentially with the number of parameters being measured, thus making the data analysis impractical[11,12].

The increasing use of single-cell technologies, especially single-cell RNA-sequencing, fostered the development of novel computational approaches that have been subsequently adapted for the unbiased investigation of FCM data[10,13,14]. A plethora of computational methods have been released in the recent years to perform single tasks such as the capacity to mark and remove outlier events (FlowAI, FlowClean, PeacoQC)[15–17], to identify cell populations through clustering (PhenoGraph, FlowSOM, SPADE1, X-shift, FlowMeans and FlowPeaks, just to mention a few)[18–22], to perform dimensionality reduction (t-SNE, UMAP) and batch correction (CytoNorm, cyCombine)[23–26]. However, these packages often lack a user-friendly graphical interface and deep knowledge of programming languages is generally required. In addition, large experiments comprising millions of cells (e.g., from immunomonitoring) are becoming progressively

[1]Laboratory of Translational Immunology, IRCCS Humanitas Research Hospital, via Manzoni 56, 20089 Rozzano, Milan, Italy. [2]Institute of Genetic and Biomedical Research, UoS Milan, National Research Council, via Manzoni 56, 20089 Rozzano, Milan, Italy. [3]Institute for Biomedical Technologies, National Research Council, via Amendola 122/D, 70126 Bari, Italy. [4]Flow Cytometry Core, IRCCS Humanitas Research Hospital, via Manzoni 56, 20089 Rozzano, Milan, Italy. [5]Present address: School of Biological Sciences, Department of Molecular Biology, University of California San Diego, San Diego, CA, USA. [6]These authors contributed equally: Simone Puccio, Giorgio Grillo. ✉e-mail: simone.puccio@humanitasresearch.it; enrico.lugli@humanitasresearch.it

more common, thus requiring computationally expensive analyses and high-performance computers. Overall, these aspects limit the broad applicability of computational tools in FCM data analysis.

To overcome these limitations, we developed ClusteRing UnsuperviSed meThods for high dimensional cYtometry data (CRUSTY), a user-friendly web tool for the rapid identification of populations in high-dimensional FCM data. CRUSTY is available through standard browsers, with an intuitive interface and user control over a wide range of parameters for each computational step. CRUSTY functionality is based on the Scanpy[27] Python package which contains the most comprehensive tool set for data analysis with the latest methodologies and frequent updates. The webtool incorporates functionalities from several clustering and visualization tools. For example, CRUSTY integrates clustering functions from PhenoGraph[28], pyVIA[29], and FlowSOM[18] packages, a nonlinear dimensionality reduction method such as umap-learn, and tools for quality control (FlowAI[15]) and batch correction (Scanorama[30]). In addition, an interactive interface is available for the rapid visualization of results in real time. Finally, CRUSTY automatically generates vectorized, high-quality figures for direct insertion in scientific manuscripts. CRUSTY is of easy accessibility through a dedicated web server (https://crusty.humanitas.it/), thus not requiring installation of packages on the user's computer. It is also provided with an intuitive tutorial-style user interface to guide logical navigation through the analysis and visualization steps. In principle, CRUSTY can be used to analyse any FCM and mass cytometry dataset, thus filling an existing gap between high-dimensional data generation and exploration.

## Results

### Design and Implementation
CRUSTY architecture is composed by (i) an input data web manager; (ii) a backend based on computational pipeline available at: https://github.com/luglilab/Cytophenograph; and (iii) an interactive cell browser for the visualization of the results.

Figure 1 shows the pipeline to analyse data in CRUSTY. FCM files are first pre-processed with a standard flow cytometry software for compensation, biexponential transformation, removal of unwanted events such as dead cells and identification of the cell population of interest, e.g., total CD8 + T cells. A detailed protocol describing data pre-processing and precautions that should be taken to avoid batch effects and bias in computational analyses (e.g., resulting from different experiments performed in different days) has been previously published[6]. These precautions take into account day-to-day standardization of machine performance with calibration beads, reproducibility of compensation outputs, pipetting error and lot-to-lot reagent validation, among others. As improved algorithms for batch correction are currently being evaluated, batch effect of flow cytometry data should be tested before uploading in CRUSTY. To help users with data pre-processing, instructions are also reported in form of a tutorial at https://crusty.humanitas.it/. Pre-processed data are then imported in CRUSTY as comma-separated values (CSV) files for (1) cluster discovery and dimensionality reduction and (2) interactive data exploration.

### Input data
Mandatory input files for CRUSTY execution are:
- One or multiple CSV files containing transformed intensity values exported from the flow cytometer's acquisition software.
- One info-file containing the metadata information associated with each single CSV file of the dataset.

### Pre-processing and quality control
Transformed intensity matrices from each CSV file are automatically combined into a single matrix using the concat function of the Scanpy package. In the combined expression matrix, each cell is given a

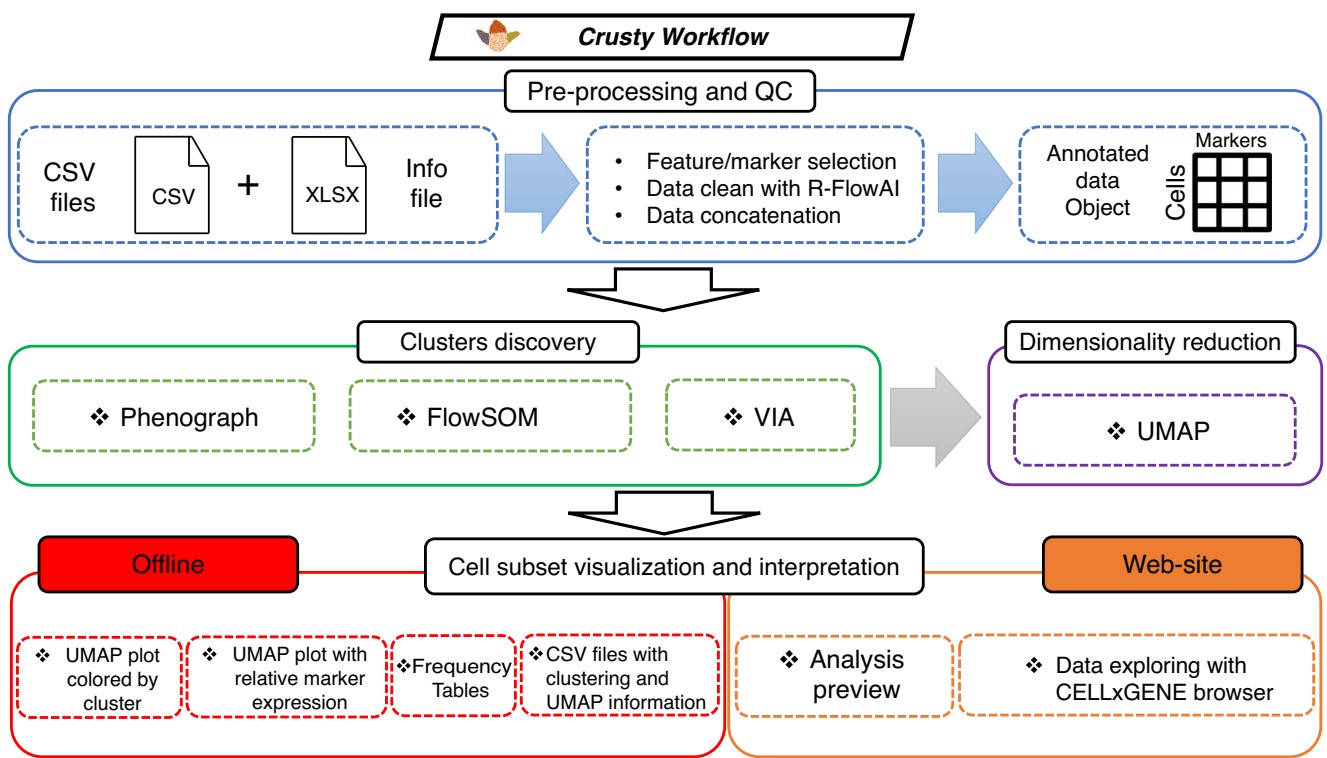

**Fig. 1 | Overview of the CRUSTY workflow.** Algorithms and data visualization types available in CRUSTY. Pre-processing (e.g., compensation, biexponential transformation) of raw FCM data are performed with standard FCM softwares prior to loading data into CRUSTY. CSV comma-separated values; UMAP Uniform Manifold Approximation and Projection.

unique ID (the combination of the original CSV file name and a progressive number starting from 0). Data cleaning is carried out on the combined matrix with the FlowAI R package[15] and consists in the detection and removal of anomalies by checking the flow rate (i.e., time parameter).

## Cluster discovery

Identification of heterogeneity within the dataset is implemented by clustering algorithms. CRUSTY is equipped with three state-of-the-art clustering methods: PhenoGraph, VIA, and FlowSOM. PhenoGraph and VIA are graph-based clustering algorithms while FlowSOM is based on self-organizing maps (SOM). The input parameter to run PhenoGraph and VIA is the K-value that represents the number of K-Nearest Neighbors for graph construction. Based on the number of neighbors shared by every two cells, PhenoGraph, calculates the similarity between cells using the Jaccard similarity coefficient that generates the adjacency matrix, which is then used to build the network. The last step is the community detection that is performed with the Leiden algorithm to extract cell communities. FlowSOM consists of three steps: (i) building a SOM, (ii) building a minimal spanning tree and (iii) computing a meta-clustering. As input requirement, FlowSOM needs the exact number of clusters the user wants to obtain. In CRUSTY, we modified the original Python code of PhenoGraph package by setting a fixed seed number, so to obtain reproducible UMAPs between different runs. Resolution to be used for cluster discovery is arbitrary and depends on the original scientific question (e.g., identification of major vs. rare cell populations). We suggest to test different levels of resolution and choose the more appropriate one for the specific use. In any case, it is strongly recommend to validate bioinformatic results with robust functional assays in vitro or in vivo. The different algorithms work at different speed and may lead to similar results depending on the resolution being used. A comparison between PhenoGraph and FlowSOM applied to high-dimensional flow cytometry data has been previously reported[6]. We suggest to refer to recent surveys for optimal choice of the algorithm to be used in specific applications[31,32].

## Dimensionality reduction

To visualize clusters, CRUSTY uses the uniform manifold approximation and projection (UMAP). In the input form under "Advanced Options", users can define the spread and the min_distance to modify the clumping of the embedded points.

## Output description

CRUSTY provides four types of output: tabular results, matrix plots, UMAP plots, and stacked bar plots showing the cluster abundance per sample and per experimental group (specified in the infofile). Tabular results include the original transformed intensity matrix with the addition of UMAP coordinates (two dimensions of the embedding) and a column with the cluster assignment for each cell in the data. CRUSTY creates two matrix plots, a first with the mean expression values per marker and a second with the scaled intensities values per marker (z-score). Tabular results are useful for further data processing with dedicated statistics. UMAP plots are generated for the rapid visualization of the data (Fig. 2). Results are stored for 2 weeks and are retrievable using a web link reported at the top of the Analysis:: Execute page which, upon request, is sent by email to the user-provided email address (optional).

To demonstrate the power of CRUSTY, we report the reanalysis of a previously published 26-color dataset comprising 36 CD8 + T cell samples from human bone marrows, lymph nodes, lungs, and peripheral blood mononuclear cells (PBMCs)[33], for a total of 180,000 cells (5,000 cells/sample). The resulting data analysis is also available on the CRUSTY website as a sample and source data can be downloaded from the CRUSTY website. Pre-processed CD8 + T cells can be downloaded at https://flowrepository.org/id/FR-FCM-Z5LE to test reproducibility.

Following analysis, in this example with VIA (totaling ~10 min in computation time), CRUSTY automatically generates several plots that are downloadable in PDF format. Figure 2 includes examples of these plots, such as tissue-specific distributions of cells in a UMAP (Fig. 2a), UMAP (Fig. 2b), and stacked bar graphs of VIA cluster distributions according to their tissue of origin (Fig. 2c), individual FCM marker distribution on the UMAP [both a tiled image showing all markers (Fig. 2d) and separate, individual markers are provided], and a heatmap showing normalized marker expressions across clusters, useful to interpret cluster identity. Several additional plots that might be useful to explore the complex nature of high-dimensional FCM datasets are also provided on the CRUSTY website.

## Cell subset visualization and interpretation

CRUSTY output is loaded into CELLxGENE, a simple and intuitive cell browser that enables the user to quickly explore results. The primary visualization is a 2D scatterplot of cells (UMAP) based on input fluorescent parameters (Fig. 3). By clicking on the "drop" function, the user can distinguish cells belonging to those categorical variables listed in the info file directly on the UMAP (a default color code is applied). By moving the cursor on variables (e.g., individual clusters or samples) listed on the left of the workspace, CRUSTY automatically highlights the selected cells directly on the UMAP. This function is particularly useful to rapidly identify those samples with exceptional variability.

In the example, we show the identification of human mucosal associated invariant T cells (MAIT), a specialized population of CD8 + T cells with innate-like functions[34], across different single-cell clusters and tissues. By using the CRUSTY Signature function, we employed the markers GZMK, CD161, and CD127 to identify MAIT cells in the dataset (Fig. 3a). CRUSTY identifies discrete levels of the signature (arbitrarily defined as low, medium, and high). Signature[high] cells are further selected and colored in the UMAP (Fig. 3a, dark blue). Signature enrichment in individual VIA clusters and tissues is shown in dedicated histograms, thereby revealing that clusters 6 and 23, and bone marrow and PBMCs have the highest MAIT signature enrichment (Fig. 3b). These results are in accordance with data presented in Fig. 2. Signature[high] cells can be further isolated for downstream analysis (Fig. 3c), such as with standard FCM bivariate plots. In the example, we show that the major difference between MAIT clusters is related to positive CD69 expression in cluster 23 (Fig. 3d), as previously revealed by VIA clustering in Fig. 2e.

## Comparison with similar tools

Although several commercial softwares are available for the analysis of FCM data, high-dimensional datasets are generally processed via open-source bioinformatics algorithm in R or Python. Web servers that are freely available for clustering analysis were recently developed but have limitations compared to CRUSTY (Table S1). Cytofkit[35] is a stand-alone software designed mainly for mass cytometry data and his release is available on Bioconductor. The Cytofkit Shiny APP is not designed for online analysis but is useful for data exploration. CytoChain[36] is a webtool developed in the form of Shiny APP but it is still under construction, does not offer an interactive interface for the data exploration in real time, and does not provide sample datasets for users training.

# Discussion

In conclusion, CRUSTY is a versatile tool for the rapid analysis and exploration of high-dimensional FCM data. It has been developed to fill a gap, i.e., to enable basic users to run complex computational algorithms without the need of bioinformatic support and knowledge of programming. Importantly, CRUSTY automatically generates publication quality figures within minutes, thereby reducing the time required from high-dimensional data exploration and draft manuscript preparation. Functions related to data pre-processing such as

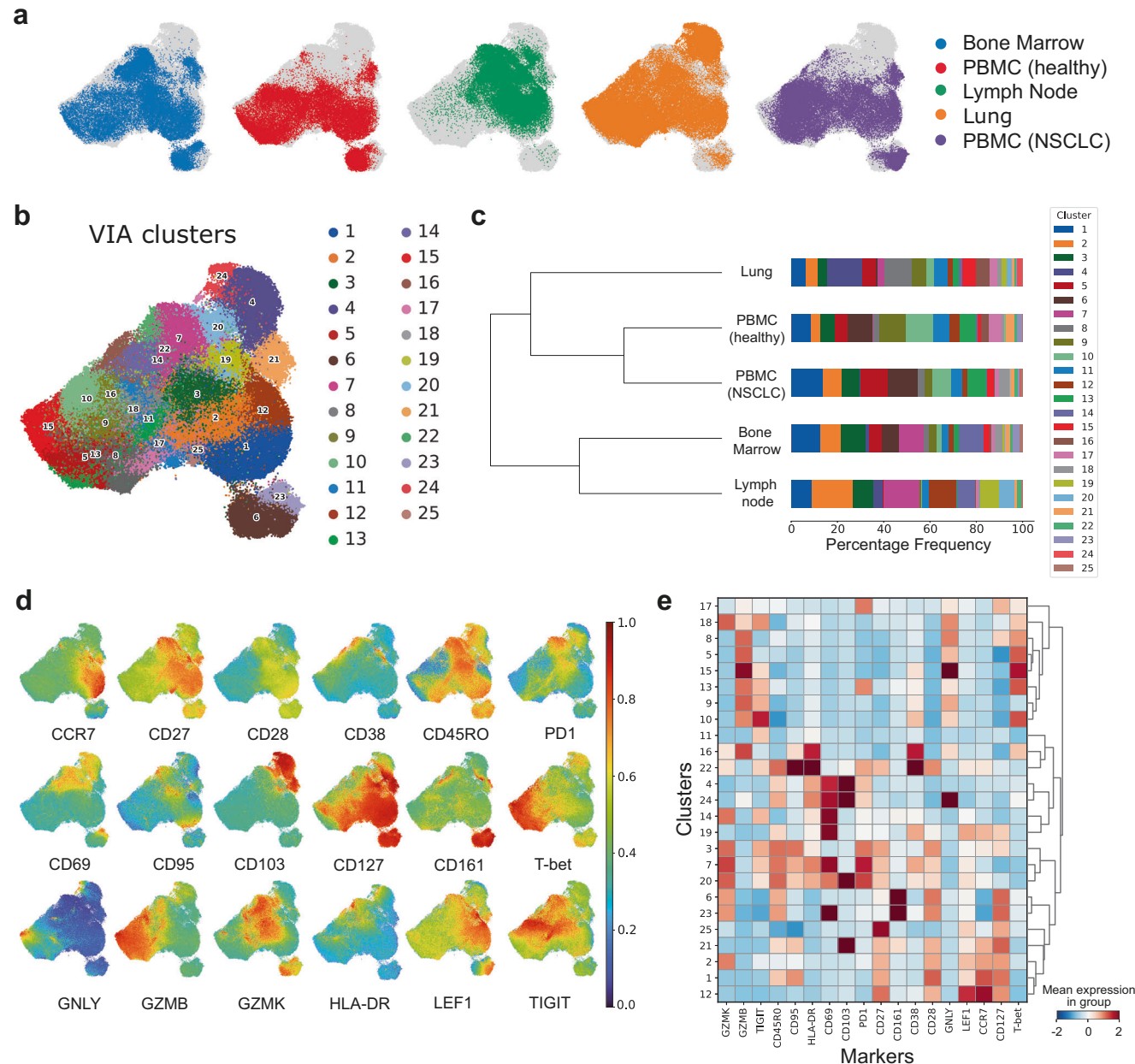

**Fig. 2 | Selected CRUSTY data outputs. a** UMAP plots showing single CD8 + T cells from specific sets of samples (here, tissues). The gray background refers to total CD8 + T cells in the dataset. UMAP distribution (**b**) and stacked-bar graphs (**c**, showing percentages) of CD8 + T cell VIA clusters. In (**c**), hierarchical clustering is performed to highlight similarities and differences between immunophenotypes in tissues. **d** Tiled UMAP plots showing the relative expression of selected markers by CD8 + T cells. **e** Heatmap of the relative expression (z-score) of markers (columns) in discrete CD8 + T cell VIA clusters (rows). Hierarchical clustering is performed to highlight similarities and differences between clusters. PBMC, peripheral blood mononuclear cells; NSCLC, non-small cell lung cancer. For graphs in (**c**) and (**e**), source data are provided as Source Data files.

compensation, biexponential transformation and gating are currently under development and are expected to be included in a future version of CRUSTY. These improvements will obviate the need of traditional flow cytometry softwares or additional platforms, and simplify data exploration even further.

## Methods
### Implementation
CRUSTY web tool implementation is based on a three-tier architecture: client, server, and database. In the client layer, the Graphical User Interface (GUI) is developed as a dynamic Web Application, built on the React web framework – release 18 - (https://reactjs.org/), the powerful Javascript library for building user interfaces, and on the popular Bootstrap 5 (https://getbootstrap.com/), the extensible and feature-packed frontend framework for responsive and mobile-first web design features. These two frameworks have allowed the creation of a complex interactive UI with a user-friendly frontend.

A specialized RESTful Web service in the server layer is responsible for asynchronous communication with the Web Interface to manage the file upload and validation, the execution monitoring, the analysis results and the interactive graphs visualization. The Django REST framework (https://www.django-rest-framework.org/), a powerful and flexible toolkit for building Web APIs, has been adopted to implement the service. The backend which includes site administration/maintenance, user management and authentication process is based on the Django high-level Python framework.

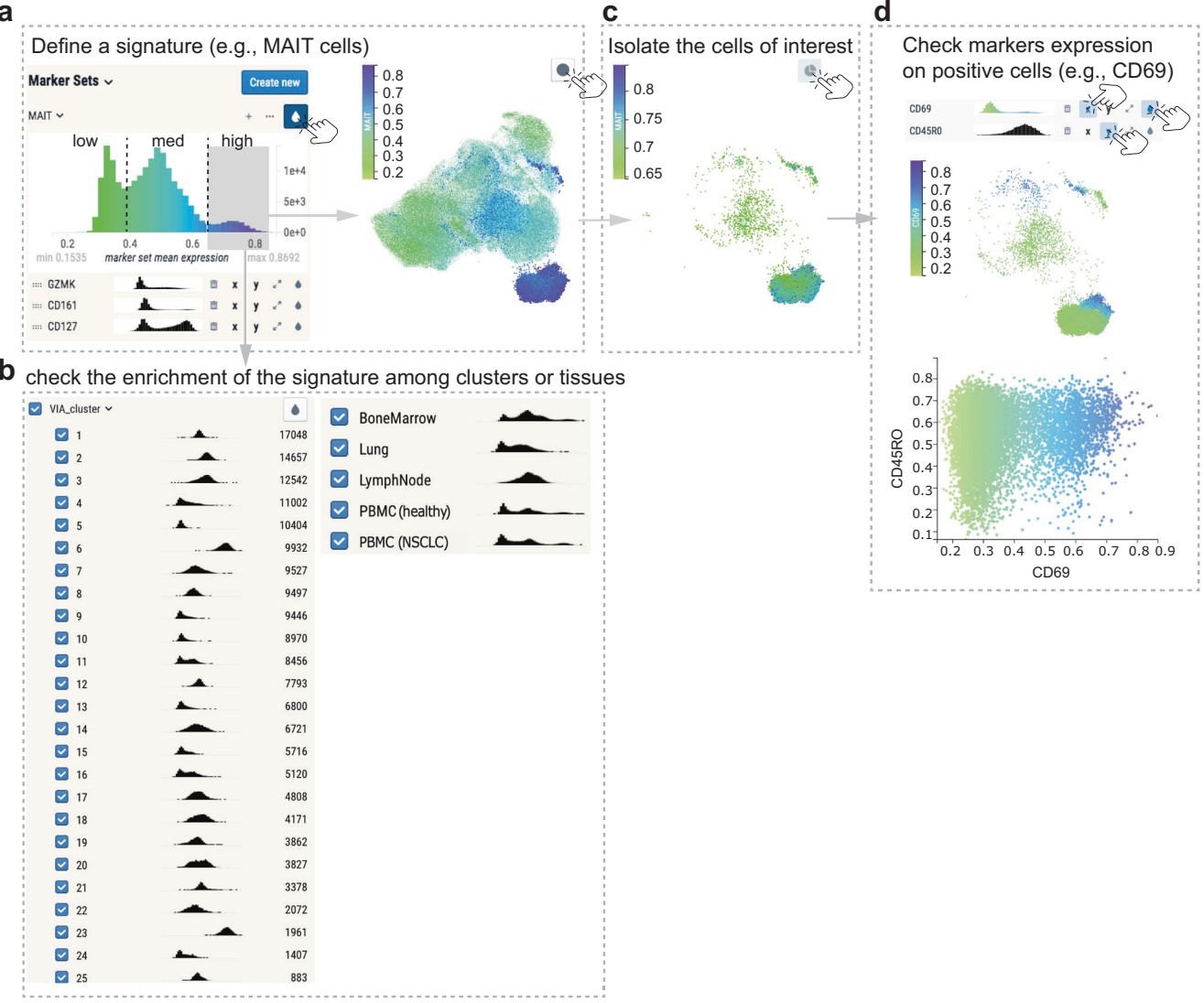

**Fig. 3 | Interactive FCM data analysis with CRUSTY. a** Definition of the MAIT signature according to GZMK, CD161, and CD127 expression, and enrichment of the signature on the UMAP plot after clicking the "drop" function (hand cursor). The signature is arbitrarily divided in low, medium (med) and high. **b** MAIT signature levels across VIA clusters and tissue specimens in the dataset. **c** Isolation of MAIT signature[high] cells from (**a**). **d** Color-coded expression of CD69 on MAIT signature[high] cells from **c** (top) and on a standard CD69 vs. CD45RO bivariate plot (bottom). In the latter, channel values were rescaled from 0 to 1. PBMC peripheral blood mononuclear cells; NSCLC non-small cell lung cancer; MAIT mucosal-associated invariant T cell.

A SQLite3 database is used to manage data analysis, usage statistics and user logging/authentication and it also interacts with the RESTful to support communications between the data layer and the Web Application.

### Session management and results availability

A universally unique identifier (UUID version 4) named Analysis ID, consisting of a unique 32-characters ID is created and associated with each analysis. It can be used to access, resume and complete the analysis and visualize or download the results at a later time. CRUSTY web tool stores the uploaded files and analysis results on the server in a user's private workspace accessible only via a RESTful web service, ensuring the user's privacy by a unique random key (Analysis ID).

### Server and browser compatibility

The Web Application is deployed on a server with 16-core CPUs (2.40 GHz), 64GB RAM and 20TB of storage under an Apache2 web server.

It is compliant with CSS3 and HTML5 standards. JavaScript libraries are implemented in ECMAScript 2018 in order to obtain high

combability with modern browsers. Thanks to the responsive design inherited by template Bootstrap, CRUSTY is also mobile and tablet friendly.

### Statistics and reproducibility

Statistical methods were not used in this paper because not relevant. To obtain reproducible UMAPs between different runs, we modified the original Python code of PhenoGraph package by setting a fixed seed number. Original files, data and codes are publicly available to reproduce data reported in this manuscript.

### Inclusion and ethics

We have followed Nature guidelines for Inclusion and Ethics in scientific research.

### Reporting summary

Further information on research design is available in the Nature Portfolio Reporting Summary linked to this article.

## Data availability

CD8 + T cell data used in this study were previously reported in Galletti et al., Fig. 1g-i[33] and were isolated as in Supplementary Fig. 1 of the same paper[33], compensated and bi-exponentially transformed with FlowJo v.10.5.0 according to standard procedures[6] and reanalyzed in CRUSTY. Pre-processed CD8 + FCM data are available at https://flowrepository.org/id/FR-FCM-Z5LE. Source data used to build Fig. 2 can be downloaded on the CRUSTY website and are provided as Source Data Files with this publication. Source data are provided with this paper.

## Code availability

Pipeline source code, implemented in Python, is freely available for download at GitHub: https://github.com/luglilab/Cytophenograph. A Docker image is available in the Docker Hub public repository at https://hub.docker.com/r/sinnamone/cytophenograph5. The image contains the deployed version of the scripts available in GitHub and the Conda environment, in order to run the pipeline on personal/private or big datasets.

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

## Acknowledgements

The authors wish to thank Diletta Di Mitri, Federica Portale, Rachele Di Donato, Paola Vella, Annalisa Coraducci, Antonino Marsala and Nicola Quadri (Humanitas), Maria Mittelbrunn (Instituto 12 de Octubre, Madrid), Rahul Roychoudhuri (University of Cambridge, UK), the members of the Lugli laboratory, of the Humanitas Flow Cytometry Core and of the "T cell club" for insightful comments, and Nicola Losito (Institute of

Biomedical Technologies, National Research Council, Bari) for server management and support. E.L. is a CRI Lloyd J. Old STAR (CRI award 3914) and is supported by the Associazione Italiana per la Ricerca sul Cancro (AIRC IG 2017 – ID 20676, AIRC IG 2022 – ID 27391 and AIRC 5×1000 program UniCanVax 22757) and by intramural funding of the Humanitas Research Hospital. E.M.C.M. is supported by the Associazione Italiana per la Ricerca sul Cancro (MFAG 26471). S.P. and GGa were supported by Fellowships from the Fondazione Italiana per la Ricerca sul Cancro-Associazione Italiana per la Ricerca sul Cancro (FIRC-AIRC). The purchase of a FACSSymphony A5 was defrayed in part by a grant from the Italian Ministry of Health (Agreement 82/2015).

## Author contributions

S.P. and E.L. conceived software development; SP and G Grillo created the software; G. Grillo, A.C., and F.L. developed the web application; S.P., G.A., C.S., G. Galletti, E.M.C.M., G.D.S., and E.L. analyzed and interpreted data; S.P., G.A., and E.L. wrote the paper; all authors edited the paper.

## Competing interests

E.L. receives research grants from Bristol Myers Squibb on a topic unrelated to this manuscript and serves as a consultant for BD Biosciences. The other authors have no competing interests.
