## [Peer Review File · Nature Communications]

CRUSTY: a versatile web platform for the rapid analysis and visualization of high-dimensional flow cytometry dataReviewer #1 (Remarks to the Author):

This manuscript serves as an introduction to the webtool CRUSTY, which is an online server for flow cytometry analysis. The webtool does not provide new functionality, in that it integrates existing data processing and data visualisations tools rather than getting new packages. As such, there is no requirement on the authors to demonstrate the reproducibility and utility of individual components - these are all validated as individual published packages. What CRUSTY provides is a user-friendly interface that allows use of these tools for rapid analysis, and this task it performs well - the tool is easy to use for non-computational scientists, and will serve to generally improve the uptake of more sophisticated data analysis tools by the wider cytometry community. I would expect reasonable uptake of this tool by the community, as it lowers the barrier of use for researchers who rely on software as opposed to running packages (still a large majority in many fields, such as immunology). While a non-canonical manuscript, recognising the work that goes into communal projects such as this is an incentive for the developed of shared tools.

Is the webtool going to be open access?

Have the authors ensured that user access works globally, and that the hosting is scalable with the large increase in use that would occur with open access?

Reviewer #2 (Remarks to the Author):

Comments for the Author

The authors have developed a web application, named CRUSTY, that enables one-stop, coding-free flow cytometry clustering analysis on a web browser. This application provides an alternative to traditional gating strategies which are largely manual, time-consuming and unscalable across multiple markers by enabling users to use unsupervised clustering methods to identify groups of cells. CRUSTY is highly valuable in that it enables users without programming proficiency to perform clustering on large flow cytometry datasets, including outlier removal, dimensionality reduction, visualisation, and results exploration. The documentation and sample datasets are well prepared and can be used by biologists with limited coding skills.

Major Feedback

CRUSTY's email functionality should be checked. It successfully sends an email when I input my Gmail email address but CRUSTY doesn't send an email to either of my university email accounts (I have checked my junk email too) when a job is submitted.

It would be good to limit the UMAP plot to 50,000 cells to make visualising the clusters easier. In my experience clusters tend to merge when visualising 100s of thousands of cells in a single UMAP plot.

Is there a reason CRUSTY does not perform additional pre-processing steps such as arcsinh transformation and normalisation between samples?

It would be good to see summary information about the Data Cleaning step such as how many cells were removed per sample. This will enable users to identify if there are issues in their pre-processing steps for specific samples. A simple bar chart with the number of cells removed per sample or the percentage of cells removed per sample would be great.

When users select the FlowSOM algorithm it would be nice to have both the meta-cluster IDs and cluster IDs in the output files.

The manuscript should be updated with details of how the number of FlowSOM meta-clusters is selected from between the minimum and maximum meta-cluster selected. If this is determined using the Elbow criterion, then it would be good to see an Elbow plot.

The paper should include the strengths of each clustering method implemented in CRUSTY to allow users to identify which clustering algorithm is best suited for their purposes. This a good reference for the comparison of FlowSOM and Phenograph is <https://doi.org/10.1186/s13059-019-1917-7>

Minor Feedback

The hyperlink on page 5 should be <https://github.com/luglilab/Cytophenograph> not <https://github.com/luglilab/Cytophenograph>;

The hyperlinks <http://crusty.ba.itb.cnr.it/#support>, <https://reactjs.org/>, <https://getbootstrap.com/>

and <https://www.django-rest-framework.org/>, and <https://github.com/luglilab/Cytopenograph> should be blue and underlined like the other hyperlinks

You could potentially make CRUSTY available to technical users by making the RESTful API publicly available. Although I suppose a technical user could follow the pipeline outlined in <https://github.com/luglilab/Cytopenograph>.

To avoid confusion with the FlowSOM metaclustering I would update the Figure 2 caption from "The latter are metaclustered to highlight similarities and differences among immunophenotypes" I believe it should be "Hierarchical clustering is performed on the latter to highlight similarities and differences between the immunophenotypes of clusters".

I believe the minimum and the maximum number of metaclusters FlowSom can process are 3 and 90 respectively. This should be reflected in the min and max cluster inputs for FlowSOM in the CRUSTY web application.

Reviewer #3 (Remarks to the Author):

I don't believe CRUSTY is ever defined in the paper-- I only learned what CRUSTY means by logging in to the website.

In Abstract, line 39, suggest changing "can analyze" to "provide data from dozens.." or something along those lines. FCM can not analyze, it is a technology that generates data which can be analyzed.

In Introduction, line 61, "investigation of millions of cells from dozens of specimens"-- why is "millions of cells" important? Could be tens of millions or hundreds of thousands from dozens, many dozens, etc. The numbers here don't add anything and can actually be confusing. Recommend changing this to be less "numerical".

Same issue as line 61 comes up in Output Description, lines 166-170. The numerical approach used again here can be confusing to the reader. I assume there were 36 samples with total cell count for all 36 samples = 1.8 million, so 5 million cells per sample? Then downsized to 180,000 cells-- was this per sample or a total of 180,000 per sample ($\times 36 = 180,000$). Please make this section more clear.

Must you use FlowJo for pre-gating the cells of interest before sending to CRUSTY? Can't you use FCS Express? It would be nice if you could quickly test other FCM software for pre-gating.

Suggest: Accept after minor revision

Point-by-point response to reviewers

Authors. We are grateful to the Reviewers for the appreciation of our work and for their constructive comments. We have modified the software and the manuscript text to address the Reviewers' concerns. We believe that the quality of the webtool and of the manuscript has very much improved. We hope our report is now available for publication in *Nature Communications*.

Reviewer #1

This manuscript serves as an introduction to the webtool CRUSTY, which is an online server for flow cytometry analysis. The webtool does not provide new functionality, in that it integrates existing data processing and data visualisations tools rather than getting new packages. As such, there is no requirement on the authors to demonstrate the reproducibility and utility of individual components - these are all validated as individual published packages. What CRUSTY provides is a user-friendly interface that allows use of these tools for rapid analysis, and this task it performs well - the tool is easy to use for non-computational scientists, and will serve to generally improve the uptake of more sophisticated data analysis tools by the wider cytometry community. I would expect reasonable uptake of this tool by the community, as it lowers the barrier of use for researchers who rely on software as opposed to running packages (still a large majority in many fields, such as immunology). While a non-canonical manuscript, recognising the work that goes into communal projects such as this is an incentive for the development of shared tools.

Is the webtool going to be open access?

Authors. In the past 4 years, we have developed CRUSTY in the absence of specific funding. Our plan for the future is to continuously update the software with the latest, most reliable algorithms, and make it as independent as possible from additional flow cytometry softwares. This will involve at least two computer scientists, whose support is difficult to justify in academic grants, especially in the long term. To support this cost, we will offer competitive annual licenses for academic over private/industry users. In any case, we foresee to have basic analysis (e.g., with a limited number of samples or cells) to be performed for free on our platform. We are confident that this solution is acceptable to the Reviewer.

Have the authors ensured that user access works globally, and that the hosting is scalable with the large increase in use that would occur with open access?

Authors. The Web Application is deployed on a server with 16-core CPUs (2.40 GHz), 64GB RAM and 20TB of storage to ensure good performance for a standard level of usage requests. In order to maintain an efficient "load balancing" of the overall computational resources, procedures have been implemented both on the web and on the server to monitor users' requests. With tests conducted in house, CRUSTY successfully analysed >250,000 cells at one time and managed >200 single files. We never encountered computational issues when multiple users all over Europe tested their own files simultaneously. In any case, our plan is to migrate CRUSTY to a more efficient server soon, so to accommodate the increased demand from the scientific community.

Reviewer #2

The authors have developed a web application, named CRUSTY, that enables one-stop, coding-free flow cytometry clustering analysis on a web browser. This application provides an alternative to traditional gating strategies which are largely manual, time-consuming and unscalable across multiple markers by enabling users to use unsupervised clustering methods to identify groups of cells. CRUSTY is highly valuable in that it enables users without programming proficiency to perform clustering on large flow cytometry datasets, including outlier removal, dimensionality reduction, visualisation, and results exploration. The documentation and sample datasets are well

prepared and can be used by biologists with limited coding skills.

Major Feedback

CRUSTY's email functionality should be checked. It successfully sends an email when I input my Gmail email address but CRUSTY doesn't send an email to either of my university email accounts (I have checked my junk email too) when a job is submitted.

Authors. No anomalies have been found in sending the email. Is it possible that the Reviewer's university inhibits receipt from non-existent email addresses? Should this be the case, we have decided to replace the `Crusty@ba.itb.cnr.it` with `crustywebtool@gmail.com` address in order to be notified if emails are not delivered.

It would be good to limit the UMAP plot to 50,000 cells to make visualising the clusters easier. In my experience clusters tend to merge when visualising 100s of thousands of cells in a single UMAP plot.

Authors. We understand the concern of the Reviewer. In general, we prefer to have all cells in the visualization to avoid underrepresentation of rare clusters.

Is there a reason CRUSTY does not perform additional pre-processing steps such as arcsinh transformation and normalisation between samples?

Authors. Preprocessing steps (compensation, transformation) were not included in this version of CRUSTY as we believe researchers would prefer to visualize raw data with a standard flow cytometry software. Nevertheless, we plan to include these steps in a future version of the software. CRUSTY currently performs a batch correction function in case one should see batch effects between experiments during preliminary analyses. As related to normalization: we are certainly aware that normalization algorithms have been developed in flow cytometry, such as CytoNorm or similar. In our experience, these perform well when clear positive and negative expression of antigen is present (e.g., lineage markers or similar), less so with smeared antigen expression. We have previously shown and discussed that batch effects can be avoided, and very good reproducibility can be obtained when carefully following specific guidelines (day-to-day machine standardization, use of the same reagents over different experiments, same operator preparing the mix of antibodies etc.; Brummelman, Nat Protoc, 2019). This aspect is briefly discussed at page 5, line 107 of the manuscript.

It would be good to see summary information about the Data Cleaning step such as how many cells were removed per sample. This will enable users to identify if there are issues in their pre-processing steps for specific samples. A simple bar chart with the number of cells removed per sample or the percentage of cells removed per sample would be great.

Authors. As explained above, preprocessing is not available with CRUSTY at the moment: the user should be able to see how many events are excluded from the analysis by standard flow cytometry software. A cleaning step, along with other features, is planned to be included in a future version of CRUSTY. In most of the examples presented in the manuscript, we isolated CD8+ T cells from different organs, thus the number of events excluded from the analysis might depend on many factors, such as cell death after freezing, abundance of non-T cell populations, etc. It is unclear if the Reviewer raised additional concerns.

When users select the FlowSOM algorithm it would be nice to have both the meta-cluster IDs and cluster IDs in the output files. The manuscript should be updated with details of how the number of FlowSOM meta-clusters is selected from between the minimum and maximum meta-cluster selected. If this is determined using the Elbow criterion, then it would be good to see an Elbow plot.

Authors. Apparently, the export of meta-cluster IDs when conducting FlowSOM seems not possible without modifying the original code. We will consider to include this feature in the future. In the past, we have attempted to use the Elbow plot approach to select the “optimal” number of clusters, which unfortunately led to inconsistent results in different types of datasets. After discussion with colleagues in the field of single cell biology as part of the Human Cell Atlas consortium, we came to the conclusion that the resolution being used for clustering is highly arbitrary and depends on the original scientific question (identification of major vs. rare cell populations, for instance). In our opinion, the important message in this regard would be that computational results should be validated by functional assays, so to support predictions/conclusions obtained with bioinformatics. This is now briefly discussed at page 6 of the manuscript.

The paper should include the strengths of each clustering method implemented in CRUSTY to allow users to identify which clustering algorithm is best suited for their purposes. This a good reference for the comparison of FlowSOM and Phenograph is <https://doi.org/10.1186/s13059-019-1917-7>

Authors. We thank the reviewer for the suggestion. We are now citing manuscripts reporting detailed comparisons between tools at page 7, line 150. The paper has been updated.

Minor Feedback

The hyperlink on page 5 should

be <https://github.com/luglilab/Cytopenograph> not <https://github.com/luglilab/Cytopenograph>;

The

hyperlinks <http://crusty.ba.itb.cnr.it/#support>, <https://reactjs.org/>, <https://getbootstrap.com/> and <https://www.django-rest-framework.org/>, and <https://github.com/luglilab/Cytopenograph> should be blue and underlined like the other hyperlinks

Authors. We edited the manuscript as requested

You could potentially make CRUSTY available to technical users by making the RESTful API publicly available. Although I suppose a technical user could follow the pipeline outlined in <https://github.com/luglilab/Cytopenograph>.

Authors. This is now available at <https://crusty.ba.itb.cnr.it/#support>

To avoid confusion with the FlowSOM metaclustering, I would update the Figure 2 caption from “The latter are metaclustered to highlight similarities and differences among immunophenotypes” I believe it should be “Hierarchical clustering is performed on the latter to highlight similarities and differences between the immunophenotypes of clusters”.

Authors. We edited the manuscript as requested

I believe the minimum and the maximum number of metaclusters FlowSom can process are 3 and 90 respectively. This should be reflected in the min and max cluster inputs for FlowSOM in the CRUSTY web application.

Authors. The Reviewer is correct but we decided to have a maximum of 31 clusters that can be obtained with FlowSOM so to avoid overfragmentation of the data. Also, it would be very difficult to visualize 91 metaclusters with CRUSTY (e.g., cluster colors would be very similar and thus difficult to distinguish).

Reviewer #3

I don't believe CRUSTY is ever defined in the paper-- I only learned what CRUSTY means by logging in to the website.

Authors. The acronym CRUSTY is now defined in the Introduction

In Abstract, line 39, suggest changing "can analyze" to "provide data from dozens.." or something along those lines. FCM cannot analyze, it is a technology that generates data which can be analyzed.

Authors. The statement has been corrected

In Introduction, line 61, "investigation of millions of cells from dozens of specimens"-- why is "millions of cells" important? Could be tens of millions or hundreds of thousands from dozens , many dozens , etc. The numbers here don't add anything and can actually be confusing. Recommend changing this to be less "numerical".

Authors. The statement has been corrected

Same issue as line 61 comes up in Output Description, lines 166-170. The numerical approach used again here can be confusing to the reader. I assume there were 36 samples with total cell count for all 36 samples = 1.8 million, so 5 million cells per sample? Then downsized to 180,000 cells-- was this per sample or a total of 180,000 per sample (x 36 =180,000). Please make this section more clear.

Authors. We modified the manuscript as requested

Must you use FlowJo for pre-gating the cells of interest before sending to CRUSTY? Can't you use FCS Express? It would be nice if you could quickly test other FCM software for pre-gating.

Authors. Any flow cytometry software could be used for pregating. We do not see any problem in this regard when using other commercial softwares as long as the data are exported in the correct file format (.csv) from .fcs.

Reviewer #1 (Remarks to the Author):

The authors have satisfactorily addressed my concerns.

Reviewer #2 (Remarks to the Author):

I am pleased to see that the authors have addressed the minor feedback and some of the major feedback. However, I still feel the webserver and manuscript would benefit from addressing some of the other major feedback points.

Major Feedback:

Original Feedback: CRUSTY's email functionality should be checked. It successfully sends an email when I input my Gmail email address but CRUSTY doesn't send an email to either of my university email accounts (I have checked my junk email too) when a job is submitted.

Authors: No anomalies have been found in sending the email. Is it possible that the Reviewer's university inhibits receipt from non-existent email addresses? Should this be the case, we have decided to replace the Crusty@ba.itb.cnr.it with crustywebtool@gmail.com address in order to be notified if emails are not delivered.

Latest Feedback: Thank you for doing this. The email functionality still doesn't work using our university email addresses. Specifically, we found that all outlook-based university emails we tested do not receive the webserver emails. We have tested email addresses from 5 major universities. Can the authors please investigate this trying outlook email addresses? The authors might consider adding a warning to the webserver if they cannot fix this.

Original Feedback: Is there a reason CRUSTY does not perform additional pre-processing steps such as arcsinh transformation and normalisation between samples?

Authors: Preprocessing steps (compensation, transformation) were not included in this version of CRUSTY as we believe researchers would prefer to visualize raw data with a standard flow cytometry software. Nevertheless, we plan to include these steps in a future version of the software. CRUSTY currently performs a batch correction function in case one should see batch effects between experiments during preliminary analyses. As related to normalization: we are certainly aware that normalization algorithms have been developed in flow cytometry, such as CytoNorm or similar. In our experience, these perform well when clear positive and negative expression of antigen is present (e.g., lineage markers or similar), less so with smeared antigen expression. We have previously shown and discussed that batch effects can be avoided, and very good reproducibility can be obtained when carefully following specific guidelines (day-to-day machine standardization, use of the same reagents over different experiments, same operator preparing the mix of antibodies etc.; Brummelman, Nat Protoc, 2019). This aspect is briefly discussed at page 5, line 107 of the manuscript.

Latest Feedback: In this case, could you indicate in the manuscript that the input files must not have batch effects or should be normalised between samples and potentially cite a reference to how users can do this, such as the pre-processing steps of Melsen et al. 2020 (<https://doi.org/10.4049/jimmunol.1901530>) or a similar workflow in python? Could you also update the manuscript stating that CRUSTY accepts FCS files as well as CSVs, please?

Original Feedback: It would be good to see summary information about the Data Cleaning step such as how many cells were removed per sample. This will enable users to identify if there are issues in their preprocessing steps for specific samples. A simple bar chart with the number of cells removed per sample or the percentage of cells removed per sample would be great.

Authors. As explained above, preprocessing is not available with CRUSTY at the moment: the user should be able to see how many events are excluded from the analysis by standard flow cytometry software. A cleaning step, along with other features, is planned to be included in a future version of CRUSTY. In most of the examples presented in the manuscript, we isolated CD8+ T cells from different organs, thus the number of events excluded from the analysis might depend on many factors, such as cell death after freezing, abundance of non-T cell populations, etc. It is unclear if the Reviewer raised additional concerns.

Latest Feedback: When we said the Data Cleaning steps, we were referring to the fact that

CRUSTY performs data cleaning using the FlowAI R package. It would be good to understand how many anomalies were removed per sample to highlight any particularly bad samples that might need to be re-sampled.

Original Feedback: When users select the FlowSOM algorithm it would be nice to have both the meta-cluster IDs and cluster IDs in the output files. The manuscript should be updated with details of how the number of FlowSOM meta-clusters is selected from between the minimum and maximum meta-cluster selected. If this is determined using the Elbow criterion, then it would be good to see an Elbow plot.

Authors: Apparently, the export of meta-cluster IDs when conducting FlowSOM seems not possible without modifying the original code. We will consider to include this feature in the future. In the past, we have attempted to use the Elbow plot approach to select the "optimal" number of clusters, which unfortunately led to inconsistent results in different types of datasets. After discussion with colleagues in the field of single cell biology as part of the Human Cell Atlas consortium, we came to the conclusion that the resolution being used for clustering is highly arbitrary and depends on the original scientific question (identification of major vs. rare cell populations, for instance). In our opinion, the important message in this regard would be that computational results should be validated by functional assays, so to support predictions/conclusions obtained with bioinformatics. This is now briefly discussed at page 6 of the manuscript.

Latest Feedback: Apologies, we should have made ourselves clearer, about what we meant by cluster and meta-cluster. We believe you are currently displaying the cluster IDs (what I previously described as meta-clusters). It would also be good to display the self-organising map IDs (what I previously described as clusters). In our experience, the self-organising map IDs can reveal rare cell populations that would otherwise get clustered into larger clusters.

New Feedback to be addressed by the authors: I found that several of my CRUSTY FlowSOM analyses output less clusters than I specified as the min number of clusters (<https://crusty.ba.itb.cnr.it/execute/0312c304-2b4c-4e36-bc71-da816c8bdd42> and <https://crusty.ba.itb.cnr.it/execute/5298cac9-10c2-4108-861d-346ddfe9a053>)

Reviewer #3 (Remarks to the Author):

I thank the authors for their attentiveness to the reviewers comments. I feel all critique items have been addressed appropriately and recommend acceptance of the revised manuscript.

Point-by-point response to reviewers

Authors. We are grateful to the Reviewers for the appreciation of our work and for proposing publication. We are addressing additional technical requests from Reviewer 2 below. We hope our report is now available for publication in *Nature Communications*.

Reviewer 2

Original Feedback: CRUSTY's email functionality should be checked. It successfully sends an email when I input my Gmail email address but CRUSTY doesn't send an email to either of my university email accounts (I have checked my junk email too) when a job is submitted.

Authors: No anomalies have been found in sending the email. Is it possible that the Reviewer's university inhibits receipt from non-existent email addresses? Should this be the case, we have decided to replace the Crusty@ba.itb.cnr.it with crustywebtool@gmail.com address in order to be notified if emails are not delivered.

Latest Feedback: Thank you for doing this. The email functionality still doesn't work using our university email addresses. Specifically, we found that all outlook-based university emails we tested do not receive the webserver emails. We have tested email addresses from 5 major universities. Can the authors please investigate this trying outlook email addresses? The authors might consider adding a warning to the webserver if they cannot fix this.

Authors. We have enabled DKIM and SPF record for our email crustywebtool@crusty.ba.itb.cnr.it to avoid rejections from servers and tested it by sending several Italian institutions. Moreover, we have also inserted a warning in the web front-end to warn users of possible problems with the SMTP server, should this happen.

Original Feedback: Is there a reason CRUSTY does not perform additional pre-processing steps such as arcsinh transformation and normalisation between samples?

Authors: Preprocessing steps (compensation, transformation) were not included in this version of CRUSTY as we believe researchers would prefer to visualize raw data with a standard flow cytometry software. Nevertheless, we plan to include these steps in a future version of the software. CRUSTY currently performs a batch correction function in case one should see batch effects between experiments during preliminary analyses. As related to normalization: we are certainly aware that normalization algorithms have been developed in flow cytometry, such as CytoNorm or similar. In our experience, these perform well when clear positive and negative expression of antigen is present (e.g., lineage markers or similar), less so with smeared antigen expression. We have previously shown and discussed that batch effects can be avoided, and very good reproducibility can be obtained when carefully following specific guidelines (day-to-day machine standardization, use of the same reagents over different experiments, same operator preparing the mix of antibodies

etc.; Brummelman, Nat Protoc, 2019). This aspect is briefly discussed at page 5, line 107 of the manuscript.

Latest Feedback: In this case, could you indicate in the manuscript that the input files must not have batch effects or should be normalised between samples and potentially cite a reference to how users can do this, such as the pre-processing steps of Melsen et al. 2020 (<https://doi.org/10.4049/jimmunol.1901530>) or a similar workflow in python? Could you also update the manuscript stating that CRUSTY accepts FCS files as well as CSVs, please?

Authors. Thank you for the suggestion. As referred to in the text, major guidelines to avoid batch effects were described in our previous Nature Protocols paper (PMID: 31160786). A detailed description on how to pre-process data for analysis is also reported. This is now emphasized in the revised version of the manuscript at page 5. Please note that we are still working on the implementation to read .fcs files directly. Algorithms to correct batch effect in flow cytometry are currently being tested. We expect to include them in an upcoming new version of the software.

Original Feedback: It would be good to see summary information about the Data Cleaning step such as how many cells were removed per sample. This will enable users to identify if there are issues in their preprocessing steps for specific samples. A simple bar chart with the number of cells removed per sample or the percentage of cells removed per sample would be great.

Authors. As explained above, preprocessing is not available with CRUSTY at the moment: the user should be able to see how many events are excluded from the analysis by standard flow cytometry software. A cleaning step, along with other features, is planned to be included in a future version of CRUSTY. In most of the examples presented in the manuscript, we isolated CD8+ T cells from different organs, thus the number of events excluded from the analysis might depend on many factors, such as cell death after freezing, abundance of non-T cell populations, etc. It is unclear if the Reviewer raised additional concerns.

Latest Feedback: When we said the Data Cleaning steps, we were referring to the fact that CRUSTY performs data cleaning using the FlowAI R package. It would be good to understand how many anomalies were removed per sample to highlight any particularly bad samples that might need to be re-sampled.

Authors. We have added a bar plot showing the number of events before and after quality control with FlowAI among the output graphs. A representative example of the output is shown below for Reviewer's convenience (not shown in the manuscript). Please note that few events were removed in this case because sample cleaning was performed manually in our lab before analysis with CRUSTY.

Editorial Fig. 1. No. of flow cytometric events before and after FlowAI quality control (QC).

Original Feedback: When users select the FlowSOM algorithm it would be nice to have both the meta-cluster IDs and cluster IDs in the output files. The manuscript should be updated with details of how the number of FlowSOM meta-clusters is selected from between the minimum and maximum meta-cluster selected. If this is determined using the Elbow criterion, then it would be good to see an Elbow plot. **Authors:** Apparently, the export of meta-cluster IDs when conducting FlowSOM seems not possible without modifying the original code. We will consider to include this feature in the future. In the past, we have attempted to use the Elbow plot approach to select the “optimal” number of clusters, which unfortunately led to inconsistent results in different types of datasets. After discussion with colleagues in the field of single cell biology as part of the Human Cell Atlas consortium, we came to the conclusion that the resolution being used for clustering is highly arbitrary and depends on the original scientific question (identification of major vs. rare cell populations, for instance). In our opinion, the important message in this regard would be that computational results should be validated by functional assays, so to support predictions/conclusions obtained with bioinformatics. This is now briefly discussed at page 6 of the manuscript.

Latest Feedback: Apologies, we should have made ourselves clearer, about what we meant by cluster and metal cluster. We believe you are currently displaying the cluster IDs (what I previously described as meta-clusters). It would also be good to display the self-organising map IDs (what I previously described as clusters). In our experience, the self-organising map IDs can reveal rare cell populations that would otherwise get clustered into larger clusters.

Authors. We appreciate your suggestion and now we switch from FlowSOM python version to the FlowSOM R version, and we have implemented the changes accordingly. In the output CSV, FCS, and h5ad files, you will now find the cluster

label and self-organizing map (SOM) IDs, which can be obtained using the following functions: `GetClusters(fSOM)` and `GetMetaclusters(fSOM)`.

In addition to these updates, we have included two new plots in the UMAP folder. The first plot is the Minimal Spanning Tree (MST) plot, generated using the `PlotStars(fSOM)` function. This plot provides a visual representation of the relationships between clusters based on their proximity in the data space.

The second plot, `PlotNumbers(fSOM)`, has also been added. This plot displays the SOM IDs and their corresponding numbers, providing a clearer understanding of the clusters' composition and their arrangement within the self-organizing map.

We believe these modifications will greatly enhance the analysis process and improve the identification of rare cell populations.

New Feedback to be addressed by the authors: I found that several of my CRUSTY FlowSOM analyses output less clusters than I specified as the min number of clusters (<https://crusty.ba.itb.cnr.it/execute/0312c304-2b4c-4e36-bc71-da816c8bdd42> and <https://crusty.ba.itb.cnr.it/execute/5298cac9-10c2-4108-861d-346ddfe9a053>)

Authors. To enhance the user interface, we now ask users to indicate the exact number of clusters for meta-clustering in FlowSOM. This adjustment allows for a more precise control over the clustering process. This is specified at page 6 of the manuscript.

Reviewer #2 (Remarks to the Author):

Thank you to the authors for updating the manuscript and website. You have satisfactorily addressed my feedback.

Point-by-point response to reviewers

Reviewer #2 (Remarks to the Author):

Thank you to the authors for updating the manuscript and website. You have satisfactorily addressed my feedback.

Authors. We are grateful to the Reviewers for the appreciation of our work, for the constructive comments and for proposing publication.